# Convolutional Networks on Graphs for Learning Molecular Fingerprints

**David Duvenaud**[†]**, Dougal Maclaurin**[†]**, Jorge Aguilera-Iparraguirre**
**Rafael Gómez-Bombarelli, Timothy Hirzel, Alán Aspuru-Guzik, Ryan P. Adams**
Harvard University

## Abstract

We introduce a convolutional neural network that operates directly on graphs. These networks allow end-to-end learning of prediction pipelines whose inputs are graphs of arbitrary size and shape. The architecture we present generalizes standard molecular feature extraction methods based on circular fingerprints. We show that these data-driven features are more interpretable, and have better predictive performance on a variety of tasks.

## 1 Introduction

Recent work in materials design used neural networks to predict the properties of novel molecules by generalizing from examples. One difficulty with this task is that the input to the predictor, a molecule, can be of arbitrary size and shape. Currently, most machine learning pipelines can only handle inputs of a fixed size. The current state of the art is to use off-the-shelf fingerprint software to compute fixed-dimensional feature vectors, and use those features as inputs to a fully-connected deep neural network or other standard machine learning method. This formula was followed by [28, 3, 19]. During training, the molecular fingerprint vectors were treated as fixed.

In this paper, we replace the bottom layer of this stack – the function that computes molecular fingerprint vectors – with a differentiable neural network whose input is a graph representing the original molecule. In this graph, vertices represent individual atoms and edges represent bonds. The lower layers of this network is convolutional in the sense that the same local filter is applied to each atom and its neighborhood. After several such layers, a global pooling step combines features from all the atoms in the molecule.

These neural graph fingerprints offer several advantages over fixed fingerprints:

- **Predictive performance.** By using data adapting to the task at hand, machine-optimized fingerprints can provide substantially better predictive performance than fixed fingerprints. We show that neural graph fingerprints match or beat the predictive performance of standard fingerprints on solubility, drug efficacy, and organic photovoltaic efficiency datasets.

- **Parsimony.** Fixed fingerprints must be extremely large to encode all possible substructures without overlap. For example, [28] used a fingerprint vector of size 43,000, after having removed rarely-occurring features. Differentiable fingerprints can be optimized to encode only relevant features, reducing downstream computation and regularization requirements.

- **Interpretability.** Standard fingerprints encode each possible fragment completely distinctly, with no notion of similarity between fragments. In contrast, each feature of a neural graph fingerprint can be activated by similar but distinct molecular fragments, making the feature representation more meaningful.

---

[†]Equal contribution.

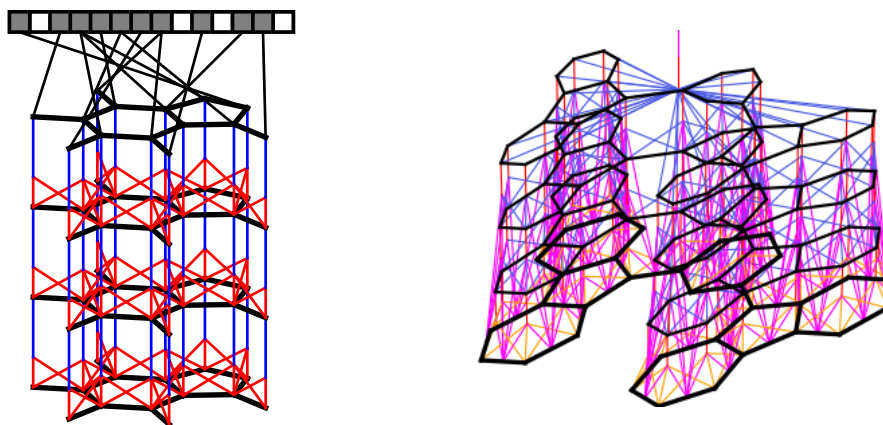

Figure 1: *Left*: A visual representation of the computational graph of both standard circular fingerprints and neural graph fingerprints. First, a graph is constructed matching the topology of the molecule being fingerprinted, in which nodes represent atoms, and edges represent bonds. At each layer, information flows between neighbors in the graph. Finally, each node in the graph turns on one bit in the fixed-length fingerprint vector. *Right*: A more detailed sketch including the bond information used in each operation.

## 2 Circular fingerprints

The state of the art in molecular fingerprints are extended-connectivity circular fingerprints (ECFP) [21]. Circular fingerprints [6] are a refinement of the Morgan algorithm [17], designed to encode which substructures are present in a molecule in a way that is invariant to atom-relabeling.

Circular fingerprints generate each layer's features by applying a fixed hash function to the concatenated features of the neighborhood in the previous layer. The results of these hashes are then treated as integer indices, where a 1 is written to the fingerprint vector at the index given by the feature vector at each node in the graph. Figure 1(left) shows a sketch of this computational architecture. Ignoring collisions, each index of the fingerprint denotes the presence of a particular substructure. The size of the substructures represented by each index depends on the depth of the network. Thus the number of layers is referred to as the 'radius' of the fingerprints.

Circular fingerprints are analogous to convolutional networks in that they apply the same operation locally everywhere, and combine information in a global pooling step.

## 3 Creating a differentiable fingerprint

The space of possible network architectures is large. In the spirit of starting from a known-good configuration, we designed a differentiable generalization of circular fingerprints. This section describes our replacement of each discrete operation in circular fingerprints with a differentiable analog.

**Hashing** The purpose of the hash functions applied at each layer of circular fingerprints is to combine information about each atom and its neighboring substructures. This ensures that any change in a fragment, no matter how small, will lead to a different fingerprint index being activated. We replace the hash operation with a single layer of a neural network. Using a smooth function allows the activations to be similar when the local molecular structure varies in unimportant ways.

**Indexing** Circular fingerprints use an indexing operation to combine all the nodes' feature vectors into a single fingerprint of the whole molecule. Each node sets a single bit of the fingerprint to one, at an index determined by the hash of its feature vector. This pooling-like operation converts an arbitrary-sized graph into a fixed-sized vector. For small molecules and a large fingerprint length, the fingerprints are always sparse. We use the `softmax` operation as a differentiable analog of indexing. In essence, each atom is asked to classify itself as belonging to a single category. The sum of all these classification label vectors produces the final fingerprint. This operation is analogous to the pooling operation in standard convolutional neural networks.

| **Algorithm 1** Circular fingerprints | **Algorithm 2** Neural graph fingerprints |
|---|---|
| 1: **Input:** molecule, radius $R$, fingerprint length $S$ | 1: **Input:** molecule, radius $R$, hidden weights $H_1^1 \ldots H_R^5$, output weights $W_1 \ldots W_R$ |
| 2: **Initialize:** fingerprint vector $\mathbf{f} \leftarrow \mathbf{0}_S$ | 2: **Initialize:** fingerprint vector $\mathbf{f} \leftarrow \mathbf{0}_S$ |
| 3: **for** each atom $a$ in molecule | 3: **for** each atom $a$ in molecule |
| 4:    $\mathbf{r}_a \leftarrow g(a)$   ▷ lookup atom features | 4:    $\mathbf{r}_a \leftarrow g(a)$   ▷ lookup atom features |
| 5: **for** $L = 1$ to $R$   ▷ for each layer | 5: **for** $L = 1$ to $R$   ▷ for each layer |
| 6:    **for** each atom $a$ in molecule | 6:    **for** each atom $a$ in molecule |
| 7:       $\mathbf{r}_1 \ldots \mathbf{r}_N = \text{neighbors}(a)$ | 7:       $\mathbf{r}_1 \ldots \mathbf{r}_N = \text{neighbors}(a)$ |
| 8:       $\mathbf{v} \leftarrow [\mathbf{r}_a, \mathbf{r}_1, \ldots, \mathbf{r}_N]$  ▷ concatenate | 8:       $\mathbf{v} \leftarrow \mathbf{r}_a + \sum_{i=1}^N \mathbf{r}_i$   ▷ sum |
| 9:       $\mathbf{r}_a \leftarrow \text{hash}(\mathbf{v})$  ▷ hash function | 9:       $\mathbf{r}_a \leftarrow \sigma(\mathbf{v} H_L^N)$  ▷ smooth function |
| 10:      $i \leftarrow \text{mod}(r_a, S)$  ▷ convert to index | 10:      $\mathbf{i} \leftarrow \text{softmax}(\mathbf{r}_a W_L)$  ▷ sparsify |
| 11:      $\mathbf{f}_i \leftarrow 1$  ▷ Write 1 at index | 11:      $\mathbf{f} \leftarrow \mathbf{f} + \mathbf{i}$  ▷ add to fingerprint |
| 12: **Return:** binary vector $\mathbf{f}$ | 12: **Return:** real-valued vector $\mathbf{f}$ |

Figure 2: Pseudocode of circular fingerprints (*left*) and neural graph fingerprints (*right*). Differences are highlighted in blue. Every non-differentiable operation is replaced with a differentiable analog.

**Canonicalization** Circular fingerprints are identical regardless of the ordering of atoms in each neighborhood. This invariance is achieved by sorting the neighboring atoms according to their features, and bond features. We experimented with this sorting scheme, and also with applying the local feature transform on all possible permutations of the local neighborhood. An alternative to canonicalization is to apply a permutation-invariant function, such as summation. In the interests of simplicity and scalability, we chose summation.

Circular fingerprints can be interpreted as a special case of neural graph fingerprints having large random weights. This is because, in the limit of large input weights, `tanh` nonlinearities approach step functions, which when concatenated form a simple hash function. Also, in the limit of large input weights, the `softmax` operator approaches a one-hot-coded `argmax` operator, which is analogous to an indexing operation.

Algorithms 1 and 2 summarize these two algorithms and highlight their differences. Given a fingerprint length $L$, and $F$ features at each layer, the parameters of neural graph fingerprints consist of a separate output weight matrix of size $F \times L$ for each layer, as well as a set of hidden-to-hidden weight matrices of size $F \times F$ at each layer, one for each possible number of bonds an atom can have (up to 5 in organic molecules).

## 4 Experiments

We ran two experiments to demonstrate that neural fingerprints with large random weights behave similarly to circular fingerprints. First, we examined whether distances between circular fingerprints were similar to distances between neural fingerprint-based distances. Figure 3 (left) shows a scatterplot of pairwise distances between circular vs. neural fingerprints. Fingerprints had length 2048, and were calculated on pairs of molecules from the solubility dataset [4]. Distance was measured using a continuous generalization of the Tanimoto (a.k.a. Jaccard) similarity measure, given by

$$\text{distance}(\mathbf{x}, \mathbf{y}) = 1 - \sum \min(x_i, y_i) \Big/ \sum \max(x_i, y_i) \tag{1}$$

There is a correlation of $r = 0.823$ between the distances. The line of points on the right of the plot shows that for some pairs of molecules, binary ECFP fingerprints have exactly zero overlap.

Second, we examined the predictive performance of neural fingerprints with large random weights vs. that of circular fingerprints. Figure 3 (right) shows average predictive performance on the solubility dataset, using linear regression on top of fingerprints. The performances of both methods follow similar curves. In contrast, the performance of neural fingerprints with small random weights follows a different curve, and is substantially better. This suggests that even with random weights, the relatively smooth activation of neural fingerprints helps generalization performance.

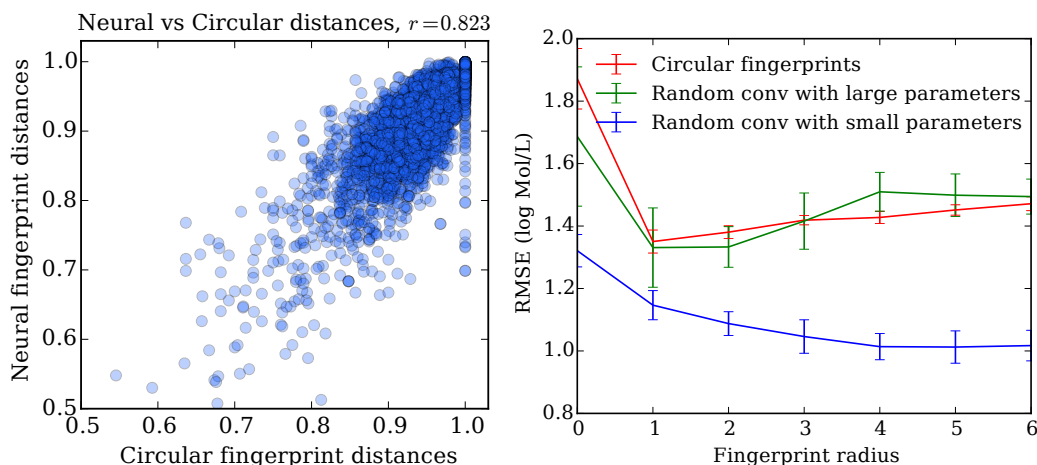

Figure 3: *Left:* Comparison of pairwise distances between molecules, measured using circular fingerprints and neural graph fingerprints with large random weights. *Right*: Predictive performance of circular fingerprints (red), neural graph fingerprints with fixed large random weights (green) and neural graph fingerprints with fixed small random weights (blue). The performance of neural graph fingerprints with large random weights closely matches the performance of circular fingerprints.

## 4.1   Examining learned features

To demonstrate that neural graph fingerprints are interpretable, we show substructures which most activate individual features in a fingerprint vector. Each feature of a circular fingerprint vector can each only be activated by a single fragment of a single radius, except for accidental collisions. In contrast, neural graph fingerprint features can be activated by variations of the same structure, making them more interpretable, and allowing shorter feature vectors.

**Solubility features**   Figure 4 shows the fragments that maximally activate the most predictive features of a fingerprint. The fingerprint network was trained as inputs to a linear model predicting solubility, as measured in [4]. The feature shown in the top row has a positive predictive relationship with solubility, and is most activated by fragments containing a hydrophilic R-OH group, a standard indicator of solubility. The feature shown in the bottom row, strongly predictive of insolubility, is activated by non-polar repeated ring structures.

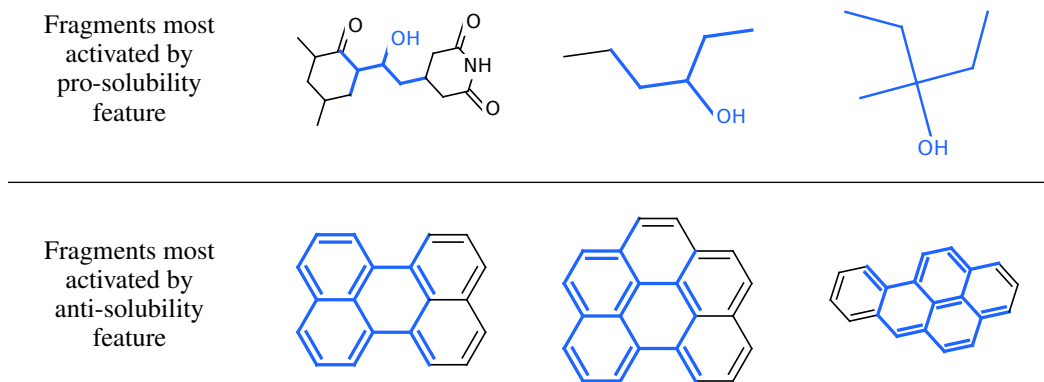

Figure 4: Examining fingerprints optimized for predicting solubility. Shown here are representative examples of molecular fragments (highlighted in blue) which most activate different features of the fingerprint. *Top row:* The feature most predictive of solubility. *Bottom row:* The feature most predictive of insolubility.

**Toxicity features**    We trained the same model architecture to predict toxicity, as measured in two different datasets in [26]. Figure 5 shows fragments which maximally activate the feature most predictive of toxicity, in two separate datasets.

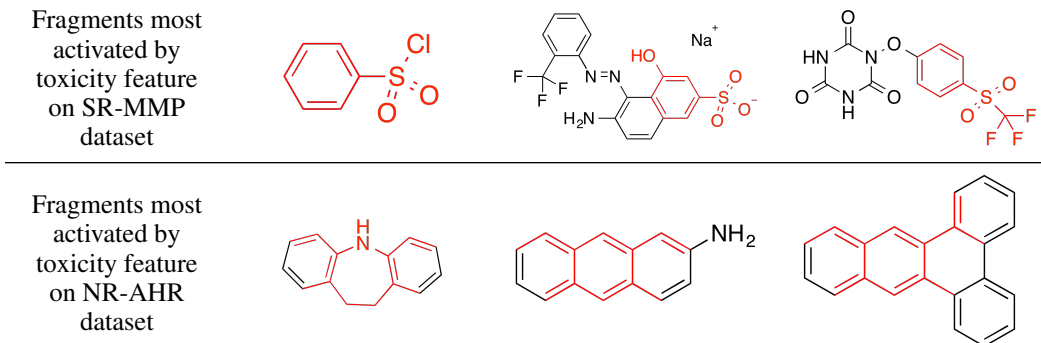

Fragments most activated by toxicity feature on SR-MMP dataset

Fragments most activated by toxicity feature on NR-AHR dataset

Figure 5: Visualizing fingerprints optimized for predicting toxicity. Shown here are representative samples of molecular fragments (highlighted in red) which most activate the feature most predictive of toxicity. *Top row:* the most predictive feature identifies groups containing a sulphur atom attached to an aromatic ring. *Bottom row:* the most predictive feature identifies fused aromatic rings, also known as polycyclic aromatic hydrocarbons, a well-known carcinogen.

[27] constructed similar visualizations, but in a semi-manual way: to determine which toxic fragments activated a given neuron, they searched over a hand-made list of toxic substructures and chose the one most correlated with a given neuron. In contrast, our visualizations are generated automatically, without the need to restrict the range of possible answers beforehand.

## 4.2   Predictive Performance

We ran several experiments to compare the predictive performance of neural graph fingerprints to that of the standard state-of-the-art setup: circular fingerprints fed into a fully-connected neural network.

**Experimental setup**    Our pipeline takes as input the SMILES [30] string encoding of each molecule, which is then converted into a graph using RDKit [20]. We also used RDKit to produce the extended circular fingerprints used in the baseline. Hydrogen atoms were treated implicitly.

In our convolutional networks, the initial atom and bond features were chosen to be similar to those used by ECFP: Initial atom features concatenated a one-hot encoding of the atom's element, its degree, the number of attached hydrogen atoms, and the implicit valence, and an aromaticity indicator. The bond features were a concatenation of whether the bond type was single, double, triple, or aromatic, whether the bond was conjugated, and whether the bond was part of a ring.

**Training and Architecture**    Training used batch normalization [11]. We also experimented with `tanh` vs `relu` activation functions for both the neural fingerprint network layers and the fully-connected network layers. `relu` had a slight but consistent performance advantage on the validation set. We also experimented with dropconnect [29], a variant of dropout in which weights are randomly set to zero instead of hidden units, but found that it led to worse validation error in general. Each experiment optimized for 10000 minibatches of size 100 using the Adam algorithm [13], a variant of RMSprop that includes momentum.

**Hyperparameter Optimization**    To optimize hyperparameters, we used random search. The hyperparameters of all methods were optimized using 50 trials for each cross-validation fold. The following hyperparameters were optimized: log learning rate, log of the initial weight scale, the log $L_2$ penalty, fingerprint length, fingerprint depth (up to 6), and the size of the hidden layer in the fully-connected network. Additionally, the size of the hidden feature vector in the convolutional neural fingerprint networks was optimized.

| Dataset | Solubility [4] | Drug efficacy [5] | Photovoltaic efficiency [8] |
|---|---|---|---|
| Units | log Mol/L | $EC_{50}$ in nM | percent |
| Predict mean | $4.29 \pm 0.40$ | $1.47 \pm 0.07$ | $6.40 \pm 0.09$ |
| Circular FPs + linear layer | $1.71 \pm 0.13$ | $\mathbf{1.13 \pm 0.03}$ | $2.63 \pm 0.09$ |
| Circular FPs + neural net | $1.40 \pm 0.13$ | $1.36 \pm 0.10$ | $2.00 \pm 0.09$ |
| Neural FPs + linear layer | $0.77 \pm 0.11$ | $\mathbf{1.15 \pm 0.02}$ | $2.58 \pm 0.18$ |
| Neural FPs + neural net | $\mathbf{0.52 \pm 0.07}$ | $\mathbf{1.16 \pm 0.03}$ | $\mathbf{1.43 \pm 0.09}$ |

Table 1: Mean predictive accuracy of neural fingerprints compared to standard circular fingerprints.

**Datasets**　We compared the performance of standard circular fingerprints against neural graph fingerprints on a variety of domains:

- **Solubility:** The aqueous solubility of 1144 molecules as measured by [4].
- **Drug efficacy:** The half-maximal effective concentration ($EC_{50}$) *in vitro* of 10,000 molecules against a sulfide-resistant strain of *P. falciparum*, the parasite that causes malaria, as measured by [5].
- **Organic photovoltaic efficiency:** The Harvard Clean Energy Project [8] uses expensive DFT simulations to estimate the photovoltaic efficiency of organic molecules. We used a subset of 20,000 molecules from this dataset.

**Predictive accuracy**　We compared the performance of circular fingerprints and neural graph fingerprints under two conditions: In the first condition, predictions were made by a linear layer using the fingerprints as input. In the second condition, predictions were made by a one-hidden-layer neural network using the fingerprints as input. In all settings, all differentiable parameters in the composed models were optimized simultaneously. Results are summarized in Table 4.2.

In all experiments, the neural graph fingerprints matched or beat the accuracy of circular fingerprints, and the methods with a neural network on top of the fingerprints typically outperformed the linear layers.

**Software**　Automatic differentiation (AD) software packages such as Theano [1] significantly speed up development time by providing gradients automatically, but can only handle limited control structures and indexing. Since we required relatively complex control flow and indexing in order to implement variants of Algorithm 2, we used a more flexible automatic differentiation package for Python called Autograd (`github.com/HIPS/autograd`). This package handles standard Numpy [18] code, and can differentiate code containing while loops, branches, and indexing.

Code for computing neural fingerprints and producing visualizations is available at `github.com/HIPS/neural-fingerprint`.

## 5　Limitations

**Computational cost**　Neural fingerprints have the same asymptotic complexity in the number of atoms and the depth of the network as circular fingerprints, but have additional terms due to the matrix multiplies necessary to transform the feature vector at each step. To be precise, computing the neural fingerprint of depth $R$, fingerprint length $L$ of a molecule with $N$ atoms using a molecular convolutional net having $F$ features at each layer costs $\mathcal{O}(RNFL + RNF^2)$. In practice, training neural networks on top of circular fingerprints usually took several minutes, while training both the fingerprints and the network on top took on the order of an hour on the larger datasets.

**Limited computation at each layer**　How complicated should we make the function that goes from one layer of the network to the next? In this paper we chose the simplest feasible architecture: a single layer of a neural network. However, it may be fruitful to apply multiple layers of nonlinearities between each message-passing step (as in [22]), or to make information preservation easier by adapting the Long Short-Term Memory [10] architecture to pass information upwards.

**Limited information propagation across the graph**    The local message-passing architecture developed in this paper scales well in the size of the graph (due to the low degree of organic molecules), but its ability to propagate information across the graph is limited by the depth of the network. This may be appropriate for small graphs such as those representing the small organic molecules used in this paper. However, in the worst case, it can take a depth $\frac{N}{2}$ network to distinguish between graphs of size $N$. To avoid this problem, [2] proposed a hierarchical clustering of graph substructures. A tree-structured network could examine the structure of the entire graph using only $\log(N)$ layers, but would require learning to parse molecules. Techniques from natural language processing [25] might be fruitfully adapted to this domain.

**Inability to distinguish stereoisomers**    Special bookkeeping is required to distinguish between stereoisomers, including enantiomers (mirror images of molecules) and *cis/trans* isomers (rotation around double bonds). Most circular fingerprint implementations have the option to make these distinctions. Neural fingerprints could be extended to be sensitive to stereoisomers, but this remains a task for future work.

# 6    Related work

This work is similar in spirit to the neural Turing machine [7], in the sense that we take an existing discrete computational architecture, and make each part differentiable in order to do gradient-based optimization.

**Neural nets for quantitative structure-activity relationship (QSAR)**    The modern standard for predicting properties of novel molecules is to compose circular fingerprints with fully-connected neural networks or other regression methods. [3] used circular fingerprints as inputs to an ensemble of neural networks, Gaussian processes, and random forests. [19] used circular fingerprints (of depth 2) as inputs to a multitask neural network, showing that multiple tasks helped performance.

**Neural graph fingerprints**    The most closely related work is [15], who build a neural network having graph-valued inputs. Their approach is to remove all cycles and build the graph into a tree structure, choosing one atom to be the root. A recursive neural network [23, 24] is then run from the leaves to the root to produce a fixed-size representation. Because a graph having $N$ nodes has $N$ possible roots, all $N$ possible graphs are constructed. The final descriptor is a sum of the representations computed by all distinct graphs. There are as many distinct graphs as there are atoms in the network. The computational cost of this method thus grows as $\mathcal{O}(F^2 N^2)$, where $F$ is the size of the feature vector and $N$ is the number of atoms, making it less suitable for large molecules.

**Convolutional neural networks**    Convolutional neural networks have been used to model images, speech, and time series [14]. However, standard convolutional architectures use a fixed computational graph, making them difficult to apply to objects of varying size or structure, such as molecules. More recently, [12] and others have developed a convolutional neural network architecture for modeling sentences of varying length.

**Neural networks on fixed graphs**    [2] introduce convolutional networks on graphs in the regime where the graph structure is fixed, and each training example differs only in having different features at the vertices of the same graph. In contrast, our networks address the situation where each training input is a different graph.

**Neural networks on input-dependent graphs**    [22] propose a neural network model for graphs having an interesting training procedure. The forward pass consists of running a message-passing scheme to equilibrium, a fact which allows the reverse-mode gradient to be computed without storing the entire forward computation. They apply their network to predicting mutagenesis of molecular compounds as well as web page rankings. [16] also propose a neural network model for graphs with a learning scheme whose inner loop optimizes not the training loss, but rather the correlation between each newly-proposed vector and the training error residual. They apply their model to a dataset of boiling points of 150 molecular compounds. Our paper builds on these ideas, with the

following differences: Our method replaces their complex training algorithms with simple gradient-based optimization, generalizes existing circular fingerprint computations, and applies these networks in the context of modern QSAR pipelines which use neural networks on top of the fingerprints to increase model capacity.

**Unrolled inference algorithms** [9] and others have noted that iterative inference procedures sometimes resemble the feedforward computation of a recurrent neural network. One natural extension of these ideas is to parameterize each inference step, and train a neural network to approximately match the output of exact inference using only a small number of iterations. The neural fingerprint, when viewed in this light, resembles an unrolled message-passing algorithm on the original graph.

## 7 Conclusion

We generalized existing hand-crafted molecular features to allow their optimization for diverse tasks. By making each operation in the feature pipeline differentiable, we can use standard neural-network training methods to scalably optimize the parameters of these neural molecular fingerprints end-to-end. We demonstrated the interpretability and predictive performance of these new fingerprints.

Data-driven features have already replaced hand-crafted features in speech recognition, machine vision, and natural-language processing. Carrying out the same task for virtual screening, drug design, and materials design is a natural next step.

#### Acknowledgments

We thank Edward Pyzer-Knapp, Jennifer Wei, and Samsung Advanced Institute of Technology for their support. This work was partially funded by NSF IIS-1421780.

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
