[Reviews · NeurIPS 2015]

Submitted by Assigned_Reviewer_1

This paper describes a method for learning vector representations of small molecules, or 'fingerprints,' using deep, convolutional neural networks.

This is high-quality work, and makes a significant contribution to the important problem of learning useful representations for chemoinformatics. It is clearly written and a pleasure to read.

In terms of originality, the neural network architecture presented here is pretty similar to the work of Lusci et al 2013. The three differences in network topology are: 1) This paper proposes the use of a different set of NN weights at each layer of the network, while Lusci ties the weights in each layer together, making the network recursive. 2) This method avoids cycles by limiting the depth of the network, compared to the approach in Lusci et al where the recursive structure rooted at each atom is forced to be a tree (cycles are broken).

3) Here, there are intermediate connections from each layer of the recursive network to the output.

But these differences, along with other technical choices (e.g. the use of the softmax activation in the output layer), can be important. The technical challenges of using methods like this can be prohibitive to people working in chemoinformatics, so relating this algorithm to the familiar technique of circular fingerprinting is valuable. There are plenty of original perspectives, ideas, and experiments in this work to warrant publication.

Specific comments:

In Figure 2, there should be a comma in line 8 between ra and r1. Also, atoms have a variable number of neighbors, and the text should mention how this is handled.

The circular fingerprints used in the paper have 2048 binary features, which have the advantage of taking up much less memory than 2048 floats. While this doesn't detract from the importance of the supervised learning results presented here, it could be a limitation for doing things like similarity searches over millions or billions of chemicals.

The size of the training and test sets were not clear. Were there 1000 training examples from each data set? 1000 minibatches of size 100?

The hyperparameters chosen by spearmint should be given, preferably in a table along with the bounds of the search space.

One major difficulty with architectures like this is that the computational benefits of doing SGD with minibatches are often lost because the NN architecture for each molecule is different. How was minibatch training done here?
Summary: A solid paper demonstrating a creative use of deep neural networks to learn molecular fingerprints for chemoinformatics.

Submitted by Assigned_Reviewer_2

The paper shows how to construct differentiable alternatives to circular fingerprints, a hash-based method for converting molecule descriptions into fixed-length vectors.

The authors substitute all steps in the original fingerprinting algorithm with differentiable equivalents, and use auto-differentiation to compute derivatives.

The paper is sound, and clearly written. The ideas are novel, learning methods are state of the art, and the auto-differentiation of such a complex algorithm shows an interesting line of applications.

I'm uncertain about the evaluation. While the authors state that circular fingerprints are state-of-the-art codes, and the pipeline is also state-of-the-art, the comparison does not show any results by other authors on the same datasets. Also, the dataset size was artificially decreased to 1000 points from the datasets each. How does not doing that affect performance?

Fig. 2: please use symbols from algorithm in main text as well. Correspondence would have helped

understand the algorithm more quickly.

Fig. 4: similar to fig. 5, explain whether what I (non-chemists) see makes "sense" from a chemists point of view.

Minor: - 216 :: space after 'the'
Summary: The paper shows how to construct differentiable alternatives to circular fingerprints, a hash-based method for converting molecule descriptions into fixed-length vectors. The evaluation seems a bit shallow, but I'm not from the field.

Submitted by Assigned_Reviewer_3

This work is related to previous work on recursive computation on graphs (Bruna et al 2013, Scarselli et al 2009, cited, but see much earlier work from Paolo Frasconi in the late 90's; the latter used straightforward SGD, as far as I remember).

The authors use state-of-the-art techniques for training deep nets (including Bayesian optimization) and mostly beat the classical preprocessing technique on predictive tasks.

It is a little bit strange to use only 1000 examples from each dataset. Deep nets strive when a lot more examples are used. The authors should use all the available data, making it also easier to compare with previously published work on these datasets. I see no good reason to limit oneself to 1000 examples when you have access to 20000.

That being said, this is an interesting application of deep nets (not completely new), worth further investigation. In terms of originality, it seems to reuse existing ideas, albeit it is not obvious what should work and what should not work on this type of data.
Summary: This paper proposes to replace previously used feature extraction for molecules based on circular fingerprints by a generalization of convolutional networks to graphs that is not very novel, although this particular application may be.

Submitted by Assigned_Reviewer_4

The authors present a novel method for turning molecular graphs into fixed-length feature vectors for use in machine learning models.

Their method replaces previously used hashing and pooling functions in the fingerprint construction pipeline with single neural network layers.

They show that circular fingerprinting is similar to their method with the network weights set to large random numbers.

They also give examples that show how similar molecular structures activate features of their fingerprints, suggesting that their method is better at encoding salient features than hashing methods.

Finally, they compare predictive accuracy on multiple benchmark datasets of their method versus current state-of-the-art, and show improvement.

This is not my domain of expertise. I may not totally understand the implications of this work in context.

That said, I will refrain from speaking to the originality or significance of this paper, but it is of high quality, and is clearly written.

Comments/questions ------------------ It's always good to hear that code will be provided.

Update: After reviewing the author rebuttal, I stand by my original review.
Summary: The authors present a novel method for turning molecular graphs into fixed-length feature vectors for use in machine learning models.

They show evidence to suggest that their method produces that are both more interpretable and produce better classification results than the state-of-the-art.

Author Feedback
Author rebuttal: We thank the reviewers for their thoughtful reviews and detailed feedback.

In response to specific comments:

Reviewer 1:
----------
We'll include more detail on how the fragment activations were visualized. To answer your question, we found the most-activating fragments for each feature by an argmax over all nodes in all molecular graphs in the training set. We then highlighted all atoms and bonds that contributed to that particular node's activation.

Reviewer 2:
----------
We agree that limiting the dataset size to 1000 affects performance, but of course we used the same dataset for both methods. We're running larger-scale experiments for the camera-ready version, and adding more baselines.

Including symbols from the algorithm box in the main text is a good suggestion. We'll incorporate it into our final draft.

As for whether the results make sense from a chemist's point of view - several of the authors of the paper are chemists, who sanity-checked the results. We'll make it clearer that the features reported by the algorithm correspond to standard indications of solubility and toxicity.

Reviewer 6:
----------
We handled the different number of bonds for each atom by using a different filter for each number of bonds. Atoms in organic molecules have up to 4 or 5 bonds max, so we trained 5 separate filters in our network . We'll clarify this point in the text.

As for floating-point fingerprints using more memory than binary fingerprints - this isn't an issue in practice, because far fewer neural fingerprints are required. To be concrete, the number of neural fingerprints chosen by the hyperparameter search was usually in the 10-100 range, while the number of binary fingerprints was usually maxed out (to 2048). We agree with your suggestion to include a table of hyperparameter ranges and the hyperparameters chosen, which should make this clear.

Minibatch training was beneficial computationally, even though each training example has a different computational graph. This is because we combined all filter applications at each layer within a minibatch into a single large matrix multiply, taking advantage of BLAS.

Reviewer 8:
----------
We will include larger-scale experiments in the camera-ready.

Regarding your point that it's not clear what will and will not work in this area - we agree! That is why we took care to generalize a standard method. Since our method generalizes this standard, it has to match or beat it (as long as we don't overfit!). A natural next step would be to systematically explore variations on this architecture.